# Benchmarking First-Principles Reaction Equilibrium Composition Prediction

**DOI:** 10.3390/molecules28093649

**Published:** 2023-04-22

**Authors:** Esteban A. Marques, Stefan De Gendt, Geoffrey Pourtois, Michiel J. van Setten

**Affiliations:** 1Department of Chemistry, KU Leuven (University of Leuven), Celestijnenlaan 200 F, 3001 Heverlee, Belgium; stefan.degendt@kuleuven.be; 2IMEC, Kapeldreef 75, 3001 Leuven, Belgium; 3ETSF European Theoretical Spectroscopy Facility, Institut de Physique, Université de Liège, Allée du 6 août 17, 4000 Liège, Belgium

**Keywords:** DFT, thermodynamics, equilibrium compositions, gas phase reactions

## Abstract

The availability of thermochemical properties allows for the prediction of the equilibrium compositions of chemical reactions. The accurate prediction of these can be crucial for the design of new chemical synthesis routes. However, for new processes, these data are generally not completely available. A solution is the use of thermochemistry calculated from first-principles methods such as Density Functional Theory (DFT). Before this can be used reliably, it needs to be systematically benchmarked. Although various studies have examined the accuracy of DFT from an energetic point of view, few studies have considered its accuracy in predicting the temperature-dependent equilibrium composition. In this work, we collected 117 molecules for which experimental thermochemical data were available. From these, we constructed 2648 reactions. These experimentally constructed reactions were then benchmarked against DFT for 6 exchange–correlation functionals and 3 quality of basis sets. We show that, in reactions that do not show temperature dependence in the equilibrium composition below 1000 K, over 90% are predicted correctly. Temperature-dependent equilibrium compositions typically demonstrate correct qualitative behavior. Lastly, we show that the errors are equally caused by errors in the vibrational spectrum and the DFT electronic ground state energy.

## 1. Introduction

Determining the thermodynamic properties of chemical reactions is of extreme importance in chemical process design and heavily relies on our ability to measure or compute Gibbs free energies [1,2]. Whilst molecular Gibbs free energies, which enable equilibrium composition predictions, may be found in literature sources, such as the NIST Chemistry WebBook [3], for simple systems, even for many relatively standard molecules, these data are not available. For new processes, such as (Area Selective) Atomic Layer Deposition ((AS)ALD) [4,5] and Chemical Vapor Deposition (CVD) [6,7,8], with experimentally difficult-to-measure and new, hypothetical molecules, the challenge is even bigger. In these cases, Density Functional Theory (DFT) [9,10] may provide an alternative source of information. Since DFT can give both the ground state electronic energy as well as the vibrational spectrum [11,12,13], it can be combined with statistical thermodynamics to yield all relevant and necessary thermodynamic quantities [14,15]. A quantity such as the equilibrium composition, which is of paramount importance in process design, is, however, rather complex; it combines many ingredients. Determining a systematic error bar is, hence, difficult, and a thorough benchmark is the only way out.

In the last 10–20 years, the number of studies that have benchmarked a specific property calculated from DFT or used a more advanced computational method has been increasing [13,16,17,18,19,20,21,22,23,24,25,26,27,28,29]. However, very few of them have focused on the complex issue of predicting the equilibrium composition systematically. Many studies have focused on the enthalpies and energies of reactions. In such studies, the accuracy of the prediction of the enthalpies and energies of reactions typically lies at around 2–10 kcal mol${}^{-1}$ [30,31,32,33,34,35,36,37,38,39,40,41] for small to medium sized organic molecules and up to 40 kcal mol${}^{-1}$ [42,43] for inorganic ones. Although enthalpies are a key component of Gibbs free energy, the effect that errors in their prediction have on the equilibrium composition has not been directly tackled. Moreover, the few studies that have explicitly dealt with equilibrium compositions have tended to focus on specific systems and have used a narrow scope [44,45]. Thus, a general and broad study quantifying the accuracy of DFT-predicted reaction equilibria is needed to broadly enable the use of DFT to guide the optimization of process conditions in many domains of chemistry.

To develop a benchmark of DFT for predicting temperature- and pressure-dependent reaction equilibria, we first need a reliable experimental reference. In this work, we collected all molecules cataloged in the Computational Chemistry Comparison and Benchmark DataBase (CCCBDB) [46] with experimental Gibbs free energy curves and elements from the first two rows of the periodic table excluding noble gases, lithium, and magnesium and including bromine. In total, this selection encompassed 117 molecules. We then calculated the Gibbs free energy of these molecules with DFT using six different exchange–correlation functionals and three basis sets of increasing size. Using these data, we calculated the equilibrium compositions for all possible independent reactions with less than 5 reagents, amounting to a total of 2648 reactions. We calculated the equilibrium temperature-dependent composition for the DFT and experimental Gibbs energies individually and directly compared the results to assess how the errors from the DFT thermodynamics affect the equilibrium composition.

## 2. Results

### 2.1. Errors for Constant Equilibrium Compositions

In our reaction set, most reactions have a fixed equilibrium composition (2164 reactions). As stated above, the signs of ΔG at the edges of the temperature range are enough to determine whether a correct composition is predicted. We present the results for this analysis in Table 1 with respect to experimental reports for the different functional and basis set combinations.

Except for the calculations of the LDA/SVP combination, DFT was able to capture the correct compositions of the reactions in more than 90% of cases. Additionally, both the functional choice as well as the basis set choice had relatively small effects on this percentage, providing marginal improvements of 4% and 8%, respectively. As expected, we observed that the performance of TZVP and QZVPP was indistinguishable. More unexpectedly, we see that hardly any improvement beyond the PBE results was made.

The errors in the molecular Gibbs free energies calculated from DFT have two origins. One is the temperature-independent part originating from the electronic total energy and the zero point vibrational energy and the other is the temperature-dependent part originating from the occupation of vibrational, translational, and rotational modes.

An important part of the error in the temperature-dependent part is associated with the harmonic approximation used to describe the vibrational modes in the enthalpy and entropy terms of the Gibbs free energy. This approximation is only valid at sufficiently low temperatures when the asymmetry of the potential energy landscape for the vibrational energy states can be neglected. To quantify the magnitude of the errors introduced by the harmonic approximation, Table 2 presents the minimal, maximal, and mean errors of the temperature-dependent part of the Gibbs free energies of all molecules for all functionals and basis sets for two temperatures, 400 and 1500 K. The temperature-dependent term of the molecules arising from the vibrational contribution differs from its experimental counterpart by −312.72 to 297.78 kJ mol${}^{-1}$ at 1500 K, while the range is only −45.53 to 60.53 kJ mol${}^{-1}$ at 400 K. Although these numbers seem huge, ultimately, we are only interested in energy differences between products and reactions. The variation over the different basis sets and functionals is minimal.

When carrying out the sign analysis approach only at lower-temperature windows [39] (300–400 K), we found an improvement of 2–4 percent points in terms of the percentage of correct predictions made. Considering only Gibbs energies between 300–400 K, the results are presented in Table 3.

The remainder of the incorrect predictions for all reactions with temperature- independent reactions from DFT are caused by ground state energy errors. We found that the set of reactions with an incorrect Gibbs reaction energy sign at a low temperature was associated with a $\Delta H_{0K,\mathrm{reaction}}$ term of around 60–150 kJ mol${}^{-1}$, which is about 10 times smaller than the average $\Delta H_{0K,\mathrm{reaction}}$, found to be above 600 kJ mol${}^{-1}$. The exception to this was the PW-LDA functional, for which no relevant threshold value was found.

### 2.2. Errors for Temperature-Dependent Equilibrium Compositions

Next, we considered the reactions that showed a change of sign in the experimental ΔG within the 0–2000 K interval. This set made up 18% of all reactions. An analysis of the sign of ΔG, was performed for these reactions as well (see Table 4 and Table 5). Again, we classified a reaction as being predicted correctly if the sign of ΔG was correctly predicted by the DFT results at the upper and lower temperature limits of the interval. In all cases, a significantly lower percentage of reactions was predicted correctly as compared to the fixed composition reactions. We observed an improvement of up to 18% if we only considered the low-temperature case in the analysis (see Table 3). Once again, the errors at 1500 K were attributed mainly to the harmonic approximation.

As previously mentioned, an erroneous sign of ΔG alone does not provide a full quantitative picture of the accuracy of the DFT predictions for reactions with changing compositions. The results for the integrated extent of reaction errors are displayed in Figure 1. It is important to note that PBE0 had the lowest integrated error on average; however, specific molecules may display better performance levels with different functionals. High absolute values indicate a high degree of prediction inaccuracy. The sign reflects changes at lower (negative) or higher (positive) temperatures as compared to the experimental cases. Interestingly, all combinations of functionals and basis sets had mean values close to 0 K, meaning that the errors had little to no preference for over- or underestimating the transition temperature. Additionally, the exchange–correlation functionals had similar behaviors in terms of spread. This is in agreement with thermodynamic values reported for some solids [47]. The standard deviations of the integrated errors for all functionals were within the range of 355 to 488 K.

In most reactions, we found that prediction errors mostly consisted of the extent of reaction change taking place at a different temperature to the experimental one. We observed that cases with a high degree of error may be qualitatively correct, i.e., with the extent of reaction evolving in the proper direction with temperature, although the inflection point, corresponding to the change in sign of ΔG, occurs at a very different temperature to the experimental one. In such cases, if the predicted extent of reaction were to be shifted by the integrated error but in the opposite direction, predictions would become far more accurate. Whilst this was the most common behavior, some reactions displayed an incorrect extent of reaction, even at low temperatures. The deviation improved as the temperature increased (see Figure 2). In these cases, DFT predicted a fixed composition over the temperature range, although a change in composition took place experimentally. Separating the cases in which this happens is a rather difficult task, as the average enthalpy (which for the case of temperature-independent equilibrium compositions was very different, and distinguishing accurate and inaccurate predictions no longer allows such distinctions) lies very close to the value needed to properly describe the reactions.

### 2.3. Error Analysis

Understanding the origin of the errors is crucial if improvements are to be attempted. In order to obtain insight into the origin of the error, we selected the case of the PBE0 functional with the QZVPP basis set and studied the error sources in more detail. We found that 2 main sources of error can be isolated, namely

Errors made on ΔH, whose dominant contribution comes from the DFT ground state energy.Errors in the transition temperature for the Gibbs free energy, which mostly come from harmonic approximation inaccuracies.

Replacing one term at a time with experimental values allows the contribution of each term’s error to the integrated error to be determined. The results for this analysis are displayed in Figure 3. Interestingly, we observed similar improvements in the interquartile range of the data in both corrections, suggesting that the two sources of errors identified are of the same importance. Another observation is that most outliers in the DFT calculations dropped down to the behavior of the remaining points when the ground state Gibbs free energy was taken from experimental values.

## 3. Methods

### 3.1. Molecule Set Collection and Reaction Generation

To build a systematic benchmark, we used the following approach. First, we conducted a search of all gas phase molecules in the Computational Chemistry Comparison and Benchmark DataBase (CCCBD) [46] containing less than 9 atoms in total and excluded all radicals and ions as well as all carbon-based cyclic compounds. In this work, we limited ourselves to small molecules; however, the inclusion of larger molecules can be done. It is known that larger molecules (>10 atoms) typically have larger errors in enthalpy [42,43], which can very significantly alter the equilibrium composition. Second, we gathered the corresponding Gibbs free energies from the NIST Chemistry WebBook [3] and discarded all molecules for which no experimental Gibbs Energy values between 300 and 1500 K were available. This selection resulted in a set of 117 molecules. From this set, we constructed a list of stoichiometrically balanced reactions containing up to four different chemical species. The reaction set contains 2648 reactions, and the distribution of the elemental composition of the reactions is shown in Figure 4. The full list of results and reactions can be found in the Appendix A.

### 3.2. Molecular Gibbs Free Energy Calculation

We constructed the atomic coordinates for every molecule in our reference set and carried out geometry optimization using Density Functional Theory (DFT) with a number of functionals and basis sets for all molecules. All DFT calculations were carried out using the Turbomole [49] package with the Exchange Correlation functionals PWLDA [50], B3-LYP [51,52], PBE [53], PBE0 [54], M06 [55], and TPSS [56], with the basis sets SVP, TZVP, and QZVPP from the def2 basis set library in Turbomole [57,58,59,60]. Whilst many Exchange Correlation (XC) functionals are available for this purpose [36,61,62], we focused on a small range of functionals without dispersion-correction (since we focused on small molecules) that spanned a reasonable part of the spectrum of computationally cheap DFT methods. We used an energy change convergence criterion of $1\times 10^{-6}$ Ha for self-consistent field convergence, an m5 grid size for the exchange–correlation integration [63], and a $1\times 10^{-5}$ Ha convergence for the convergence of the ground state. We converged the SCF calculations as well as the geometry optimization with a convergence criterion of $1\times 10^{-6}$ Ha. The geometry optimization scheme was used with a maximum of 0.3 Å coordinate changes per iteration. Further tightening of these criteria showed no effect on the equilibrium compositions in the sampled reactions. We stress that computationally cheap is of the essence, since chemical systems of reactions can quickly grow into many species whose properties need to be considered.

In the next step, we calculated the vibrational frequencies of all molecules in the harmonic approximation. From these, we obtained the vibrational partition function $q_{v}$. The rotational and translational partition functions, $q_{r}$ and $q_{t}$, were obtained in the ideal gas approximation [14,15]. Finally, the electronic contribution $q_{e}$ originating from the spin multiplicity was included. In full detail, the molecular Gibbs free energy $G_{\mathrm{molecular}}$ is given by
(1)$$G_{\mathrm{molecular}}=E_{0}+E_{\mathrm{ZPE}}+RT\ln\left(Q\right)+PV_{\mathrm{ideal}}-TS$$
where $E_{0}$ is the electronic ground state energy of the molecule, $E_{\mathrm{ZPE}}$ is the vibrational zero point energy of the molecule, *P* is the reference pressure (1 bar), $V_{ideal}$ is the ideal volume of a single molecule, T is the temperature, and Q is the full molecular partition function:(2)$$Q=q_{v}q_{r}q_{t}q_{e}$$

The vibrational contribution $q_{v}$ is given by
(3)$$q_{v}=\prod_{j}\frac{e^{-\frac{h\omega_{j}}{2k_{B}T}}}{1-e^{-\frac{h\omega_{j}}{k_{B}T}}}$$
where $k_{B}$ is the Boltzmann constant, and $\omega_{j}$ are the wave number of mode *j* calculated from the harmonic approximation. The translational part becomes
(4)$$q_{t}=V{\left(\frac{2\pi mk_{B}T}{h^{2}}\right)}^{3/2}$$
with *m* being the mass of the molecule, *h* being the Plank constant, and *V* being the molecular volume, which can be expressed as follows under the ideal gas approximation:(5)$$q_{t}=\frac{k_{B}T}{P}{\left(\frac{2\pi mk_{B}T}{h^{2}}\right)}^{3/2}$$
where *P* is the reference pressure of the system (chosen as 1 atmosphere). The rotational part becomes
(6)$$q_{r}\approx \frac{\sqrt{\pi }}{\sigma }\frac{{\left(k_{B}T\right)}^{3}}{\Theta_{A}\Theta_{B}\Theta_{C}}$$
where $\Theta_{A}$, $\Theta_{B}$, and $\Theta_{C}$ are rotational constants given by the following formula:(7)$$\Theta_{i}=\frac{h^{2}}{8\pi^{2}I_{i}k}$$
where $I_{i}$ is the moment of inertia in coordinate *i* Finally, the electronic part is given by
(8)$$q_{e}=\sum_{j}g_{j}e^{-\frac{\epsilon_{j}}{k_{B}T}}$$
where $g_{j}$ is the degeneracy of the electronic mode *j* and $\omega_{j}$ is the energy of mode *j* calculated from DFT.

From the complete molecular partition function *Q*, we calculated the molecular Gibbs free energy $G_{\mathrm{molecular}}$ according to Equation (Equation 1) at equidistant temperature points in the range of 0 K–2000 K and fit a 6th order polynomial. Storage of the polynomial coefficient allowed us to readily calculate Gibbs free energies for any temperature within that range. In all cases, the fitting error was recorded and had a Root Mean Square Error (RMSE) value of 0.048 kJ mol${}^{-1}$, and the highest deviation computed was 0.28 kJ mol${}^{-1}$.

### 3.3. Calculation of the Equilibrium Composition

The equilibrium composition for any given reaction system is reached when the Gibbs free energy reaches a minimum. In order to obtain it, we used the Sequential Least-Squares Quadratic Programming (SLSQP) [64] algorithm to evolve the composition at a fixed pressure and temperature from a stoichiometric amount of reactants until a convergence criterion was reached ($1\times 10^{-10}$% of the Gibbs energy of the initial composition). We then repeated this calculation in equidistant steps spanning 300–1500 K at 1 atmosphere with entropy mixing conducted using ideal mixing laws for gases, as shown in Equations (Equation 9) and (Equation 10)).
(9)$$\Delta G_{\mathrm{mix}}=-T\Delta S_{\mathrm{mix}}$$
with
(10)$$\Delta S_{\mathrm{mix}}=-NR\sum_{i}x_{i}ln\left(x_{i}\right)$$
where *i* is the index for all species in the reaction system, *x*${}_{i}$ is the molar fraction of the component, *N* is the total amount of moles in the reaction system, and *R* is the gas constant. Minimization was performed under particle number conservation conditions.

### 3.4. Error Calculation

Finally, we needed a clear way to quantify the accuracy of the equilibrium composition prediction. Among the studied reactions we observed two distinct patterns, reactions with one fixed composition over the entire temperature range and those that included a transition. Due to mixing entropy, this distinction is not completely black and white; a partial transition can also occur. To make the distinction strict, we considered the sign of the Gibbs free energy of the reaction at 0 and 2000 K. If it was the same, we classified the reaction as a fixed composition type and if it changed, we classified it as a changing type. For the first class, the DFT-based prediction was considered either correct or incorrect. For the second class, we needed a method to quantify the error. In total, we obtained 580 reactions that included a change in the considered temperature interval.

A change in equilibrium means that the Gibbs free energy of the reaction changes signs at some point in the temperature range. The error in the calculated value of this temperature could be taken as a quantitative measure of the quality of the prediction. An example of a reaction in which this occurs is the BCl + BCl${}_{3}$→ B${}_{2}$Cl${}_{4}$ reaction, as shown in Figure 5. For this system, the DFT results predicted the transition temperature at 1300 K, whereas the experimental data placed it at 1600 K. Despite the 300 K difference in temperature, the qualitative comparison does not seem too different. To provide an alternative measure, we also considered the integrated difference in the extent of reaction (see Figure 6). This analysis was performed for all 580 reactions that showed a composition change.

## 4. Conclusions

In this work, we showed that thermodynamic data obtained from DFT calculations can be used to correctly predict equilibrium compositions from temperatures ranging from 300 to 1500 K in a broad variety of chemical reactions. Predictions for which no change in equilibrium composition is present are particularly accurate, whilst predictions that involve an equilibrium composition change are more difficult to capture but still provide a qualitative picture. This trend can still hold true for complex reaction systems (i.e., systems with multiple extents of reactions) since the qualitative nature of the predictions is likely to hold as long as $\Delta H_{0K,\mathrm{reaction}}$ is above the threshold of 50–150 kJ mol${}^{-1}$. Whether these trends still hold for molecules beyond 9 atoms requires further study.

In the set of reactions with a fixed equilibrium composition, the functionals TPSS and PBE0 at the QZVPP basis set performed the best, achieving correct predictions in almost all cases (92%), with a significant portion of the incorrect results occurring at high temperatures at which deviations from the harmonic approximation are known to dominate. Importantly, we observed that the difference in accuracy between the TZVP and QZVPP results was very small. The significant additional computational burden of the quadruple zeta basis set is, hence, hardly justifiable. Similarly, we observed a significant difference between the results obtained with the LDA functional and all others, but no real improvement was made beyond this. The incorrect reactions were all found to have a Gibbs free energy of around 50 kJ mol${}^{-1}$, i.e., 6 times lower than the average ΔG, which could be a useful threshold to consider when carrying out similar calculations.

In the set of reactions with temperature-dependent compositions, the initial error calculations showed that the sign of ΔG for this system can be wrong at the limits of the selected temperature ranges in 35–40% of all cases. In up to half of the reactions for which an incorrect sign was predicted, ΔG was incorrect at high temperatures. Despite this, qualitative information may still be extracted. Further analysis by measuring the integrated extent of reaction error showed that the composition error consists of a shift in the reaction of the extent with an RMSE of 200–300 K. These deviations were found to arise from a combination of errors in the ground state energy of the species in the reaction as well as errors in vibrational frequencies. It was shown that whilst both corrections are significant, both must be corrected in order to significantly improve the results.

## Figures and Tables

**Figure 1 molecules-28-03649-f001:**
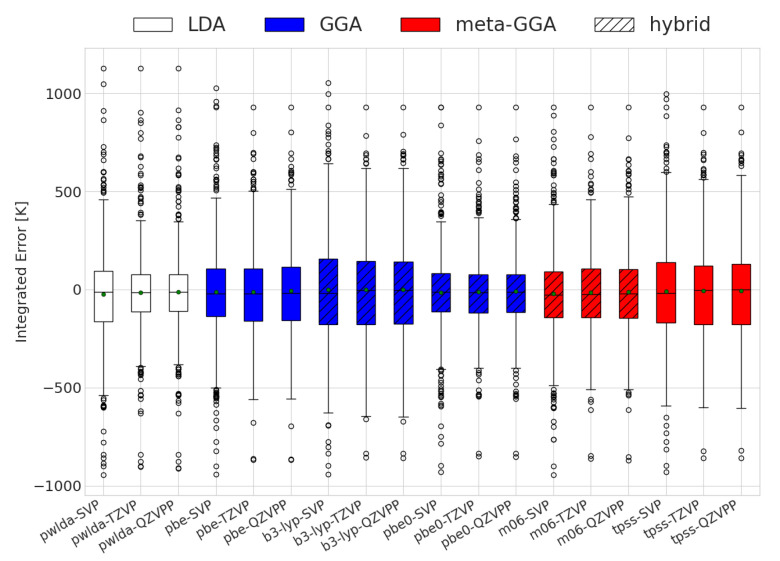
Evaluation of the integrated differences between the DFT and experimentally calculated equilibrium compositions for the different functional and basis sets used. The white, red, and blue colors depict the different exchange and correlation functionals used.

**Figure 2 molecules-28-03649-f002:**
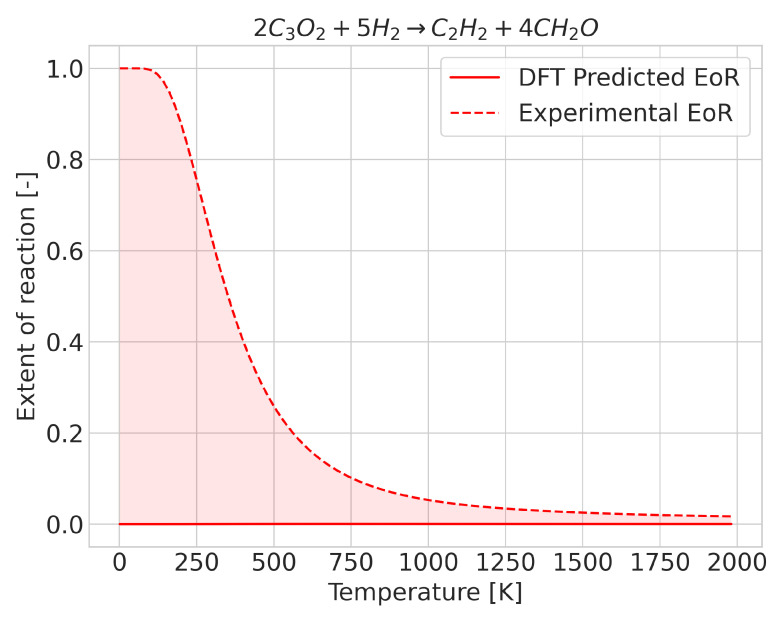
Extent of reaction for the reaction $2C_{3}O_{2}+5H_{2}\to C_{2}H_{2}+4CH_{2}O$ where the enthalpy of reaction has an incorrect sign (at 0 K).

**Figure 3 molecules-28-03649-f003:**
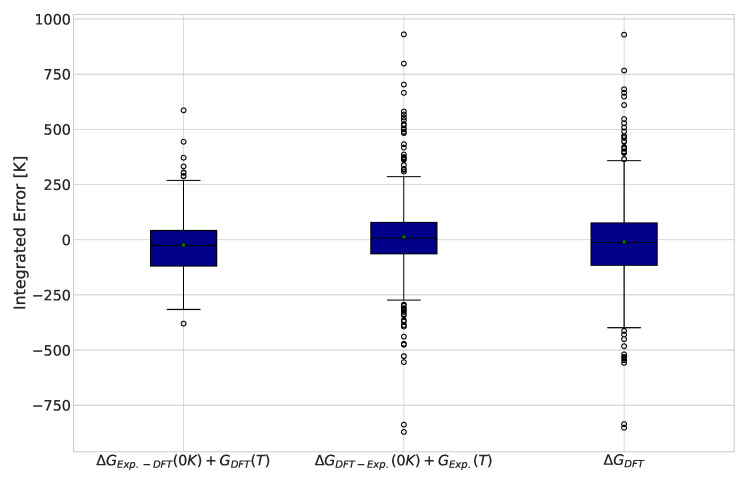
Integrated differences between DFT and experimentally calculated equilibrium compositions for 484 reactions. Green markers indicate the average error for each functional and basis set combination.

**Figure 4 molecules-28-03649-f004:**
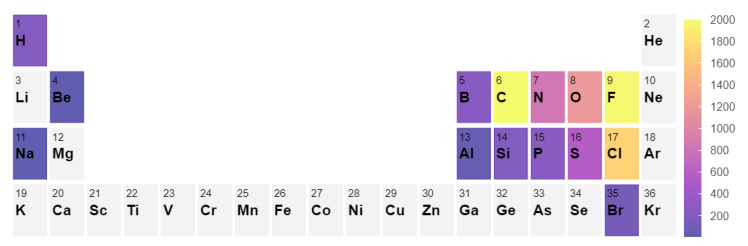
Element count in the 2648 reaction set considered in the benchmark [48].

**Figure 5 molecules-28-03649-f005:**
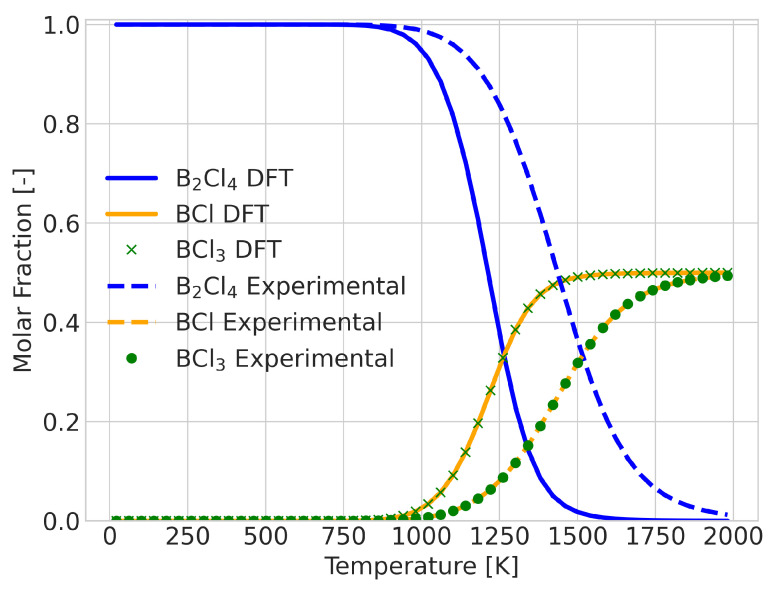
Reaction composition for the reaction BCl + BCl${}_{3}$→ B${}_{2}$Cl${}_{4}$ with incorrect signs for the Gibbs energy of the reaction between 1300 and 1600 K, showing good qualitative information.

**Figure 6 molecules-28-03649-f006:**
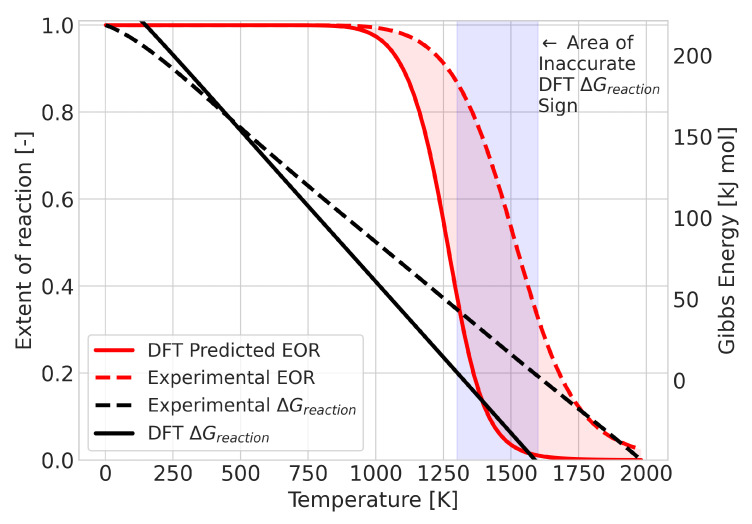
Graphical representation of the integrated difference (area shaded in red) between the experimental and DFT results for the reaction BCl + BCl${}_{3}$→ B${}_{2}$Cl${}_{4}$.

**Table 1 molecules-28-03649-t001:** Percentage of all reactions with a constant equilibrium composition correctly described by DFT with respect to the experimental reports.

Percentage of Correctly Described Reactions (%)						
	**Functional**					
**Basis Set**	**PWLDA**	**PBE**	**B3-LYP**	**PBE0**	**M06**	**TPSS**
SVP	88.7	92.1	92.7	92.8	92.6	91.5
TZVP	90.5	94.2	94.7	94.8	94.4	95.1
QZVPP	90.7	94.6	94.8	95.2	94.5	94.9

**Table 2 molecules-28-03649-t002:** DFT minimum, maximum, and mean errors of the Gibbs free energy of all reactions with respect to experimental values and all combinations of basis sets and exchange–correlation functionals given at 1500 K and 400 K.

Temperature		Error at 1500 K [Kj mol${}^{-1}$]			Error at 400 K [Kj mol${}^{-1}$]		
**Functional**	**Basis Set**	**Max**	**Min**	**Mean**	**Max**	**Min**	**Mean**
b3-lyp	QZVPP	296.21	−309.67	60.94	60.19	−45.08	11.07
b3-lyp	SVP	295.18	−310.7	59.9	60.05	−45.22	10.93
b3-lyp	TZVP	296.26	−309.62	60.98	60.20	−45.07	11.08
m06	QZVPP	295.07	−310.81	59.8	60.03	−45.25	10.91
m06	SVP	293.16	−312.72	57.89	59.74	−45.53	10.62
m06	TZVP	294.96	−310.91	59.69	60.01	−45.27	10.89
pbe0	QZVPP	295.65	−310.23	60.38	60.12	−45.15	11.00
pbe0	SVP	294.96	−310.92	59.69	60.05	−45.23	10.93
pbe0	TZVP	295.72	−310.16	60.44	60.14	−45.14	11.02
pbe	QZVPP	298.07	−307.81	62.79	60.52	−44.76	11.40
pbe	SVP	297.16	−308.72	61.88	60.41	−44.86	11.29
pbe	TZVP	298.2	−307.68	62.92	60.54	−44.73	11.42
pwlda	QZVPP	297.26	−308.62	61.99	60.44	−44.84	11.32
pwlda	SVP	296.12	−309.76	60.84	60.28	−44.99	11.16
pwlda	TZVP	297.43	−308.45	62.15	60.47	−44.80	11.35
tpss	QZVPP	297.7	−308.18	62.42	60.46	−44.81	11.34
tpss	SVP	296.79	−309.09	61.52	60.35	−44.92	11.23
tpss	TZVP	297.78	−308.1	62.5	60.48	−44.79	11.36

**Table 3 molecules-28-03649-t003:** Percentage of all reactions with a constant equilibrium composition correctly described by DFT in the range of 300 K–400 K.

Percentage of Correctly Described Reactions (%)						
	**Functional**					
**Basis Set**	**PWLDA**	**PBE**	**B3-LYP**	**PBE0**	**M06**	**TPSS**
SVP	89.1	93.1	93.5	94.5	93.6	93.5
TZVP	91.3	95.3	95.9	96.7	95.9	96.1
QZVPP	91.4	95.9	96.0	96.9	96.0	96.2

**Table 4 molecules-28-03649-t004:** Percentage of all reactions with the temperature-dependent equilibrium composition correctly described by DFT in the temperature range of 300 K–1500 K.

Percentage of Correctly Described Reactions (%)						
	**Functional**					
**Basis Set**	**PWLDA**	**PBE**	**B3-LYP**	**PBE0**	**M06**	**TPSS**
SVP	65.9	71.3	72.9	71.5	71.3	74.0
TZVP	69.2	73.6	76.2	74.4	74.4	75.0
QZVPP	69.4	73.3	76.4	74.0	74.0	74.6

**Table 5 molecules-28-03649-t005:** Percentage of all reactions with temperature-dependent equilibrium compositions correctly described by DFT in the temperature range of 300 K–400 K.

Percentage of Correctly Described Reactions (%)						
	**Functional**					
**Basis Set**	**PWLDA**	**PBE**	**B3-LYP**	**PBE0**	**M06**	**TPSS**
SVP	85.5	83.5	87.0	82.4	83.7	85.7
TZVP	88.2	85.1	87.8	82.9	84.7	85.7
QZVPP	88.2	84.7	87.6	82.9	84.5	85.3

## Data Availability

All the data produced for this work can be found in the Appendix A.

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
