# Peer review of "Benchmarking First-Principles Reaction Equilibrium Composition Prediction"

_molecules, 2023, doi:10.3390/molecules28093649_

Round 1

Reviewer 1 Report

The authors presented a systematic work of benchmarking the performance of six XC functionals and three basis sets in DFT calculations for predicting the equilibrium compositions of molecular reactions. This manuscript provides useful information to the field and a solid foundation for future relevant work. It can be published after minor revisions. Below are some comments for the authors:

1. The authors used the same convergence criteria for all the species, functionals, and basis sets. Did all the molecules used in the DFT calculations have the same convergence criteria? Did the authors test them?

2. The hybrid and meta-GGA functionals were expected to give more accurate results than PBE, but the actual difference shown in the manuscript is quite small. Is this conclusion valid for all molecules? Or are there still some specific molecules that show relatively large differences?

3. Among all the studied molecules, did the authors notice any patterns, which kinds of molecules could be better predicted by PBE, and which could be better predicted by hybrid or meta-GGA functionals?

The language is generally good, with only a few misused words and missed punctuation.

Reviewer 2 Report

  This manuscript deals mainly with benchmarking first principles reaction equilibrium composition prediction. 117 molecules were collected in this work with the experimental thermochemical data available, and 2648 reactions were constructed. These data were adopted to benchmark DFT for 6 exchange-correlation functionals and 3 qualities of basis sets. It is found that over 90% reactions without temperature dependence in the equilibrium composition below 1000 K were correctly predicted. And temperature-dependent equilibrium compositions typically demonstrate the correct qualitative behavior. In addition, the errors were found equally caused by the errors from the vibrational spectra and the DFT electronic ground state energies. 

   The authors of this manuscript carried out interesting DFT calculations for the reaction equilibrium composition prediction involving 117 small molecules and 2648 reactions, and the results would be helpful for the exploration of benchmarking reaction equilibrium composition prediction. This manuscript was largely well written and clearly organized. So this article could be recommended for publication in Molecules. In addition, there are several points that may be considered by the authors. 

1) The calculations in this manuscript include 117 small molecules containing less than 9 atoms. A further analysis, at least in one or two sentences, for the calculations of larger molecules would be useful. 

2) In line 98 of page 3, “molecular the full partition function” may be changed to “molecular full partition function”. 

3) In line 187 of page 7, “from only considering Gibbs energy” can be changed to “by considering only Gibbs energy”.

4) In the section of Conclusion, the applicability of present calculations may be further illustrated, e.g., whether or not this method could be adopted for more complex molecules and reactions.

The authors are advised to check the text for the grammar imperfections. 
